# Health Disparities of Employees in Taiwan with Major Cancer Diagnosis from 2004 to 2015: A Nation- and Population-Based Analysis

**DOI:** 10.3390/ijerph16111982

**Published:** 2019-06-04

**Authors:** Ya-Yuan Hsu, Chyi-Huey Bai, Chung-Ching Wang, Wei-Liang Chen, Wei-Te Wu, Ching-Huang Lai

**Affiliations:** 1Division of Labor Market, Institute of Labor, Occupational Safety, and Health, Ministry of Labor, Taipei 221, Taiwan; syy@mail.ilosh.gov.tw; 2School of Public Health, College of Public Health, Taipei Medical University, Taipei 110, Taiwan; 3Department of Public Health, College of Medicine, Taipei Medical University, Taipei 110, Taiwan; 4Division of Family Medicine, Department of Family and Community Medicine, Tri-Service General Hospital, National Defense Medical Center, Taipei 114, Taiwan; bigching@gmail.com (C.-C.W.); weiliang0508@gmail.com (W.-L.C.); 5National Institute of Environmental Health Sciences, National Health Research Institutes, Miaoli 350, Taiwan; 010224@nhri.org.tw; 6School of Public Health, National Defense Medical Center, Taipei 114, Taiwan; lgh@mail.ndmctsgh.edu.tw

**Keywords:** health disparities, labor insurance database, health insurance research database, cancer screening program, first-time cancer diagnosis, direct standardized incidence rates

## Abstract

*Background*: Health disparities related to environmental exposure exist in different industries. Cancer is currently a leading cause of morbidity and mortality worldwide. Much remains unknown about the types of work and industries that face the greatest cancer risks. In this study, we aimed to provide the overall and specific cancer incidences among all workers from 2004 to 2015. We also aimed to show the all-cause mortality for all employees with a first-ever cancer diagnosis. *Methods*: All workers in Taiwan in the labor insurance database in 2004–2015 were linked to the national health insurance databases. The annual overall and specific cancer incidences in 2004–2015 were calculated and stratified by industry and gender. Age-standardized incidence rates were also calculated. *Results*: A total of 332,575 workers (46.5% male) who had a first-ever cancer diagnosis from 2004–2015 were identified from 16,720,631 employees who provided 1,564,593 person-years of observation. The fishing, wholesale, construction, and building industries were identified as high-risk industries, with at least 5% of employees within them receiving a first-ever cancer diagnosis. Temporal trends of cancer incidences showed a range from 235.5 to 294.4 per 100,000 with an overall upward trend and an increase of 1.3-fold from 2004 to 2015. There were significant increases over that time for breast cancer (25%); colon cancer (8%); lung, bronchial, and tracheal cancers (11%); and oral cancer (1.7%). However, the incidence rates of cervical cancer and liver and intrahepatic cholangiocarcinoma decreased by 11.2% and 8.3%, respectively. Among the 332,575 workers with a first-ever cancer diagnosis, there were 110,692 deaths and a mortality rate of 70.75 per 1000 person-years. *Conclusions*: The overall incidence of cancer increased over the 10-year study period, probably due to the aging of the working population. High-risk industries are concentrated in the labor-intensive blue-collar class, which is related to aging and socioeconomic status intergradation.

## 1. Introduction

Cancer is a leading cause of morbidity and mortality worldwide. The World Health Organization (WHO) estimated that about 9.6 million cancer deaths occurred in 2018. Globally, about one in six deaths is due to cancer. The diagnosis of new cancer cases in underdeveloped countries is projected to increase from 56% of the total cases worldwide in 2008 to more than 60% in 2030 [1]. Meanwhile, the total annual economic cost of cancer in 2010 was estimated to be approximately US$ 1.16 trillion [2].

A number of high-income countries, such as those in North America and Western Europe, as well as Japan, South Korea, Australia, and New Zealand, have reported that lung, colorectal, breast, and prostate cancers have the highest incidence rates. Meanwhile, intermediate rates were recorded for stomach, liver, esophageal, and cervical cancers in Canada, South America, Eastern Europe, and many South Asian countries [3]. About 55% of the burden of all diseases in developed countries is associated with cancer [4]. The International Labor Office (ILO) and European Union (EU) provided recommendations for promoting the prevention of occupational cancers [5,6]. The WHO has advocated for well-established national cancer control programs [7]. Taiwan has been engaged in ongoing efforts to prevent cancer. In 2015, more than 1.81 million people accepted free adult cancer screening preventive health check services [8].

In Taiwan, based on the standards of the International Agency for Research on Cancer (IARC), the Ministry of Labor has been engaged in the classification of 52 occupational carcinogens, including lung cancer, skin cancer, blood cancer, bone cancer, liver cancer, urinary tract cancer, etc. [9,10]. Due to the long incubation period that is typical for occupational cancers, it is difficult to collect complete occupational exposure data regarding various potential causes and related confounders [11].

In the present study, a retrospective cohort study design was used. This study aimed to estimate the first-ever overall and specific cancer incidence rates among all insured employees in various industries from 2004 to 2015 in Taiwan. It also aimed to express the all causes mortality and job transition in workers with first-ever cancer diagnosis.

## 2. Methods

### 2.1. Data Sources

In this study, the nation- and population-based databases in Taiwan were used. This included the Labor Insurance Database (LID) which is maintained by the Taiwan Bureau of Labor Insurance, Ministry of Labor, as well as health-related databases—the National Health Insurance (NHI) Database (NHID), the Taiwan Cancer Registry (TCR), and the death database—which are maintained by the Ministry of Health and Welfare, Taiwan [12,13]. All work-related information for all employees, including each employee‘s industry, employment data, work years, and salaries are included in the LID. The NHID covers all diagnoses and treatments conducted by hospitals for outpatients and inpatients. The NHID, TCR, and death database contain registration files and original claims data indicating the specific diseases of and medical treatments received by the beneficiaries, as well as any deaths. Approximately 99% of workers in Taiwan participate in the LID and NHI programs. The TCR captures 97% of cancer cases in Taiwan.

### 2.2. Study Design and Study Populations

All insured workers in the LID from 2004 to 2015 were used in this study as the study population. They were linked to the NHID from 2007 to 2015 and the TCR from 1979 to 2015. All employees (*n* = 16,720,631) from the LID had links to the TCR, and 332,575 new cases of cancer were identified. In our study, a dynamic cohort was used to calculate the annual cancer incidence, and a fixed cohort was used for the analysis of the death rate and job transition rate.

In the dynamic cohort, all employees insured for at least 30 days in the index year (insured date is the index date) were included. They were linked to the NHID and TCR to identify cancer onset. New diagnostic cases were identified as incident events. All cases confirmed before the index date were excluded. There were 9,349,804 persons in 2004; 9,664,029 persons in 2005; 9,770,072 persons in 2006; 9,872,730 persons in 2007; 9,977,845 persons in 2008; 9,991,467 persons in 2009; 10,338,043 persons in 2010; 10,684,619 persons in 2011; 10,891,399 persons in 2012; 10,891,583 persons in 2013; 11,004,073 persons in 2014; and 11,187,062 persons in 2015 at risk.

In the fixed cohort, all employees with a first-ever cancer diagnosis during 2004 to 2015 were included. All confirmed cancer cases for 1979–2003 were excluded from the cohort. From 2004–2015, 332,575 new cancer cases were identified from all employees (*n* = 16,720,631) according to the oncology codes in TCR. They were followed until 2015 to identify the death dates and causes. Then, to express the 5-year survival rate in insured workers with first-ever cancer diagnosis, we also excluded 14,090 persons who did not return to work after first being diagnosed with cancer, 16,416 persons who began working after first being diagnosed with cancer, and 144,013 workers who were first diagnosed with cancer after 2011. Finally, the remaining 158,056 workers who had their first-ever diagnosis of cancer were enrolled and followed until 2015 to identify any deaths and job changes.

### 2.3. Definitions of Industry and Diagnosed Cancer

The industry codes of employees were based on the 9th revision of the industry distribution system of medium classification, which was published by the Directorate—General of Budget, Accounting and Statistic, Executive Yuan, Taiwan [14]. Cancer types were classified according to the International Classification of Diseases, Ninth Revision, Clinical Modification (ICD-9-CM) or the ICD-10-CM from the NHID or TCR. The primary sites (topography) and histologies (morphologies) of the malignancies were identified and coded according to the International Classification of Diseases for Oncology 3rd Edition (ICDO-3) published by the WHO in 2000 [15]. Cases with a behavior code of 2 or 3 in the ICDO-3 were included in TCR.

### 2.4. Data Analysis

Demographic data (e.g., gender and age), industry, and working years were gathered from LID, and cancer types, the initial diagnosis date, the cancer severity, related treatments, death date, and death causes were gathered from NHID, TCR, and the death database, and these factors were included in the analysis. SAS statistical package version 9.3 (SAS, Cary, NC, USA) was used for data management and statistical analysis.

The annual total cancer incidence rates were calculated as the number of new events divided by the total number of insured persons (at risk population) in the index year (shown in Table 1). Annual specific cancer incidence rates were also calculated as new event numbers of cancer types divided by the same denominator. The annual total and specific cancer incidences were also calculated for all industries and in both sexes. The death rate was also calculated as the total number of deaths divided by the total number of newly diagnosed workers.

For age adjustment, the age direct standardized incidence rates (SIR) were used. First, the population was grouped into 10-year age groups. Then, the age-specific incidence rates of workers were calculated. The World Standard Population (WHO 2000–2025) was used as the reference population. Then, the SIR was stratified by the cancer type, industry, sex, and year of diagnosis.

### 2.5. Ethics

Ethical approval was obtained from the Tri-Service General Hospital, National Defense Medical Center, Taipei, Taiwan Joint Institutional Review Board (Approval No.: 1-107-05-129).

## 3. Results

At first, we showed the annual total and specific cancer incidences based on a dynamic cohort. The annual cancer incidence rate in each year from 2004–2015 is shown in Table 1. The annual incidence of first-ever cancer diagnosis among insured workers from 2004–2015 ranged from 235.5 (per 100,000 people) to 294.0 (per 100,000 people) with an increase of 58.5 (per 100,000 people).

Table 2 shows the numbers of diagnoses of the top ten cancer types among all employees from 2004 to 2015. The cancer cases diagnosed from 2004 to 2015 included breast cancer (59,658 cases); liver and intrahepatic bile duct cancer (33,852 cases); cervical cancer (29,828 cases); oral cancer (29,046 cases); lung, bronchial, and tracheal cancers (23,956 cases); colon cancer (22,560 cases); rectal and anal cancers (16,378 cases); thyroid cancer (15,715 cases); leukemia (11,858 cases); and stomach cancer (9764 cases). The annual specific cancer incidences of top ten cancer types are also shown in Table 2.

The temporal trend of annual specific incidences of the top seven cancer types among all employees are shown in Figure 1. Accounting for more than 77% of all cancers in Taiwan, the seven major cancers include breast cancer; liver and intrahepatic cholangiocarcinoma; cervical cancer; oral cancer; lung, bronchial, and tracheal cancers; colon cancer; and rectal and anal cancers. Breast cancer had the most marked increase in incidence (around 25%) of all types of cancers during the 12-year period, with a significant increase of about 1.7-fold. The incidences of liver and intrahepatic bile duct cancer and cervical cancer decreased, while those of oral cancer and lung, bronchial, and tracheal cancers presented rising trends. The rates of colon cancer; rectal and anal cancer; lung, bronchial, and tracheal cancers; and oral cancer also increased (by 11%, 5%, 8%, and 1.7%, respectively).

Table 3 shows the age direct standardized incidence rates of first-ever cancer diagnosis among employees in the top 30 industries. The obvious high-risk industries were the sand, stone, and clay industry; the petroleum and coal products manufacturing industry; the real estate development industry; the chemical material manufacturing industry; and the fisheries industry. However, there were different findings between sexes. For men, the highest ranked industries were the sand, stone, and clay industry; the petroleum and coal products manufacturing industry; the real estate development industry; the fisheries industry; and the construction industry. For women, the highest ranked industries were the sand, stone, and clay industry; the human resources and supply industry; the pharmaceutical and medical chemical manufacturing industry; the legal and accounting services industry; and the creative and artistic performance industry, which first became one of the highest-ranking industries for first-ever cancer diagnosis in 2010.

Then, we carried out an investigation of a fixed cohort of all employees with a new onset first-ever cancer diagnosis in 2004–2015. A total of 332,575 employees (46.5% males and 53.5% females) with a first-ever cancer diagnosis were registered in 2004–2015 in Taiwan. Table 4 shows the basic characteristics among the studied population. The largest percentage of these patients was aged 50–54 years (20.7%), followed by those aged 55–59 years (18.2%), and the average age of diagnosis was 49.7 ± 10.2 years. A total of 110,692 persons from the 332,575 workers (1,564,593 person-years) died in 2004–2015. The mortality rate was therefore 70.75 per 1000 person-years. The proportion of workers with a first-ever cancer diagnosis who died in less than 2 years was 21.6%, and the age of such workers was also most concentrated at 55–59 years (7.3%) and 50–54 years (6.6%).

Then, to express the 5 year survival and job change rates, we analyzed the remaining 158,056 first-ever cancer workers in Table 5 after excluding those persons who did not return to work after first being diagnosed with cancer, those who began to work after first being diagnosed with cancer, and those who were first diagnosed with cancer after 2011. Table 5 shows that a total of 71.5% of workers with a first-ever cancer diagnosis had an adjustment or change to their salary at the same company, and 96.4% returned to their original company to work. The average number of days it took for those workers to return to work was 117 days, while 19.3% were re-employed by different companies. Only 93,549 (59.2%) workers were still alive at 5 years after diagnosis. The average number of days taken for re-employment was 95 days. The average salary level of the cancer patients was NT$ 27,730.

## 4. Discussion

As far as we know, this is the first longitudinal study to provide the first-ever overall and specific cancer incidences among all Taiwanese employees, as well as mortality rates among the first-ever cancer workers in Taiwan, which, in this study, were based on the national labor insured databases and linked to the health insurance and cancer registration databases in Taiwan. The study showed that fewer male than female employees had a first-ever cancer diagnosis, with 22,022 versus 32,891 cases. The annual first-ever cancer incidences of workers from 2004–2015 ranged from 235.5 (per 100,000 people) to 294.0 (per 100,000 people), with an increase of 58.5. Our results are similar to those found for Eastern Mediterranean Region (EMR) countries from 2005 to 2015, where the average incidence of cancer cases per 100,000 people increased by 46.1 [16].

Breast cancer is the most common cancer in women. More than 1.3 million patients suffering from this type of cancer are diagnosed each year around the world. A prior study has documented that this is a common malignancy in Asian women, with about 639,824 cases of breast cancer having been recorded in 2012. In that study, the number of cases among workers in Taiwan was lower than in China (the highest numbers), India, Japan, Indonesia, Pakistan, Israel, Lebanon, Armenia, Singapore, and Kazakhstan [4]. In addition, our observations in workers showed an incidence of 34 per 100,000 persons, which was lower than the incidence shown in Ferlay’s study (54 per 100,000). In some Asian countries, such as Taiwan, Hong Kong, South Korea, and Thailand, effective screening programs have been implemented [17]. Since 2005, Taiwanese women over 45 years of age have been recommended to have mammograms once every 2 years [18]. This might be one of the reasons for the increasing trend for breast cancer diagnosis.

A total of 854,000 cases of liver cancer were reported worldwide in 2015 [19]. Countries that previously had high rates of liver cancer, such as Taiwan, China, and Japan, have shown decreasing trends due to reduced aflatoxin exposure and human papillomavirus (HBV) infection rates [20,21]. After the HBV vaccination was introduced in Taiwan in 1984 [22], the incidence rate of liver and intrahepatic cholangiocarcinoma decreased by 8.3% over the next 30 years.

An estimated 526,000 cervical cancer cases occurred in 2015 worldwide, with the total incidence rate having decreased by 26% from 2005 to 2015 [23]. This is largely because the HPV vaccination has been shown to reduce cervical cancer rates in high-income countries [24,25]. Since 2011 in Taiwan, a national HPV vaccination program has been implemented based on a recommendation of the WHO made in 2008 [26]. Our data show that the five-year relative incidence for cervical cancer also decreased by 3.5% from 2011 to 2015.

Oral cancer is most likely to be associated with tobacco use and heavy alcohol use and often occurs in older male populations [27]. In this study, oral cancer maintained its position as the fourth most commonly occurring cancer among workers in Taiwan. Lung, bronchial, and tracheal cancers are also caused by tobacco use and certain occupational exposures as well as air pollution, both indoor pollution from cooking, heating, or asbestos, and outdoor pollution from particulate matter [28,29,30]. Tobacco and alcohol use are still the most important risk factors in blue-collar workers [6]. Taiwan first established smoke prevention and control legislation based on WHO policy in 1997 [31,32]. Despite this, our results indicated an increase in the incidence of lung, bronchial, and tracheal cancers of around 1.52-fold from 2004 to 2015.

Around the world, there were 1.7 million cases of colon and rectal cancer in 2015, and such cancer cases increased by 37% (32.1%–41.0%), from 1.2 million to 1.7 million cases, between 2005 and 2015 [33]. Rates in Asia have been increasing, and the increases observed are potentially due to lifestyle [3,34,35]. In this study, the temporal trends for colon cancer and rectal and anal cancer showed respective increases of 11% and 5%. The increasing trend did not slow in Taiwan, probably because colon and rectal cancer screening has been implemented since 2004 [36].

Despite the disruption of screening services during our observation period, employees in different industries were still recorded in the same health care system. Therefore, the comparisons between industries are still meaningful. Regarding occupational cancer risks, our findings were consistent with previous studies, indicating high rates of first-ever cancer in industries involving chemicals, petrochemicals, manufacturing, waste disposal and treatment, cement, power generation, mining, and metals [30,37,38]. These workers are exposed to repairable crystalline silica, diesel and gasoline engine exhaust, and various solvents. Higher SIRs of gasoline-related lung and liver cancers were identified among Taiwanese male workers in the sand, stone, and clay industry; the petroleum and coal products manufacturing industry; and the construction industry [39,40]. Stocks et al. found that male construction industry workers in the UK had an increased risk for respiratory diseases associated with exposure to asbestos, polycyclic aromatic hydrocarbons (PAHs), and welding fumes [41]. Still, other studies have found that frequent exposure to various chemical agents and ionizing radiation is related to breast and cervical cancers in women. Higher cancer rates were also identified in female Taiwanese workers in the sand, stone, and clay industry, as well as in the pharmaceutical and medical chemical manufacturing industry [42,43]. Our results were similar to these studies.

Socioeconomic status is also a determinant of health and illness. Similar to women in Taiwan, female workers in the human resources and supply industry in Japan have been found to have the highest risk of cancer, perhaps because they are often temporary workers who have many periods of unemployment, on-call duty, or work and family factors that may entail psychosocial risks. Relatedly, this study found that first-ever cancer diagnoses showed an average length of follow-up for survivors of 6 years and that these survivors had an average salary of NT$ 27,730 (around 880 USD).

The strength of this study is its large sample size. The population-based national LID and NHID cover over 99% of people in Taiwan. We provided sufficient power to detect the most current cancer incidence data for Taiwan. More people diagnosed by cancer type were shown by using the ICDO-3 to identify cancer primary sites (topography) and histologies (morphologies). In addition, retrospective cohort study analyses can provide longitudinal priori evidence evaluation following established criteria and procedures.

However, the limitations of this study should also be considered. First, our observations did not include uninsured workers and workers who left the workforce after cancer diagnosis. This study may have a potential bias caused by uninsured workers, who represent 1% of the working population in Taiwan. Second, career-related burdens are a better indicator of health differences. However, our database does not include disease-related medical expenses. Labor insurance provides a fixed amount to support cancer treatment. There is no difference in these data between workers. Therefore, we studied disease onset and mortality as the main outcomes. Finally, we used the national database, LID, NHID, and TCR, which are implemented by the government. Therefore, the causal inferences might be complicated because inferences regarding environmental monitoring and occupational exposure cannot be made with certainty.

## 5. Conclusions

At present, the IARC’s Global Initiative for Cancer Registry Development has been established and has already issued global cancer incidence data for five continents for 2008 to 2012. In Taiwan, the incidence of overall cancer has increased in the last 10 years, probably due to the aging of the working population. Regarding first-ever cancer incidence rates in all employees in Taiwan, we provide temporal trends from 2004–2015 as a baseline to discuss possible occupational cancer risks. High-risk industries are concentrated in the labor-intensive blue-collar class, which is related to aging and socioeconomic status intergradation. Further studies should consider occupation exposure in a more precise way.

## Figures and Tables

**Figure 1 ijerph-16-01982-f001:**
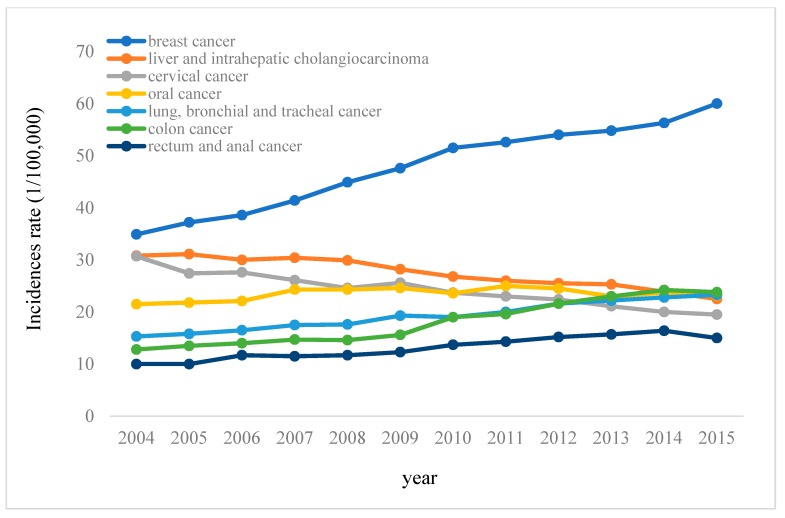
Temporal trends in annual specific first-ever cancer incidence rates for the top seven cancer types among insured workers in Taiwan, 2004–2015.

**Table 1 ijerph-16-01982-t001:** Overall incidence of first-time cancer diagnosis among insured workers in Taiwan, 2004–2015.

	2004	2005	2006	2007	2008	2009	2010	2011	2012	2013	2014	2015
Total number of insured workers	9,457,954	9,783,103	9,898,130	10,011,140	10,128,100	10,146,570	10,504,820	10,865,401	11,086,233	11,090,412	11,209,374	11,403,577
Number of individuals with first-time cancer diagnosis	22,022	22,867	23,857	25,271	26,161	26,777	28,269	29,800	31,343	31,223	32,094	32,891
Number of individuals diagnosed with cancer before this year	108,150	119,074	128,058	138,410	150,255	155,103	166,777	180,782	194,834	198,829	205,301	216,515
Number of individuals without cancer	9,327,782	9,641,162	9,746,215	9,847,459	9,951,684	9,964,690	10,309,774	10,654,819	10,860,056	10,860,360	10,971,979	11,154,171
Cancer incidence (per 10^5^)	235.5	236.6	244.2	256	262.2	268	273.4	278.9	287.8	286.7	291.7	294

**Table 2 ijerph-16-01982-t002:** Annual incidence of first-ever cancer diagnosis of the top 10 cancer types among insured workers in Taiwan, 2004–2015.

Primary Site	Total Events	2004		2005		2006		2007		2008		2009		2010		2011		2012		2013		2014		2015	
		*N*	*I*	*N*	*I*	*N*	*I*	*N*	*I*	*N*	*I*	*N*	*I*	*N*	*I*	*N*	*I*	*N*	*I*	*N*	*I*	*N*	*I*	*N*	*I*
Breast	59,658	3264	34.9	3597	37.2	3770	38.6	4086	41.4	4478	44.9	4755	47.6	5320	51.5	5625	52.6	5884	54	5968	54.8	6200	56.3	6711	60
Liver and intrahepatic bile duct	33,852	2884	30.8	3009	31.1	2928	30	3001	30.4	2979	29.9	2818	28.2	2769	26.8	2780	26	2778	25.5	2753	25.3	2632	23.9	2521	22.5
Cervical	29,828	2869	30.7	2647	27.4	2697	27.6	2576	26.1	2457	24.6	2553	25.6	2453	23.7	2453	23	2439	22.4	2299	21.1	2199	20	2186	19.5
Oral	29,046	2014	21.5	2102	21.8	2158	22.1	2399	24.3	2423	24.3	2456	24.6	2444	23.6	2674	25	2669	24.5	2500	23	2610	23.7	2597	23.2
Lung, bronchial, tracheal	23,956	1430	15.3	1524	15.8	1610	16.5	1727	17.5	1761	17.6	1925	19.3	1960	19	2134	20	2350	21.6	2415	22.2	2508	22.8	2612	23.3
Colon	22,560	1200	12.8	1300	13.5	1367	14	1451	14.7	1460	14.6	1554	15.6	1962	19	2093	19.6	2348	21.6	2508	23	2658	24.2	2659	23.8
Rectum and anus	16,378	933	10	970	10	1140	11.7	1136	11.5	1172	11.7	1231	12.3	1417	13.7	1529	14.3	1653	15.2	1706	15.7	1810	16.4	1681	15
Thyroid	15,715	760	8.1	789	8.2	852	8.7	1004	10.2	1102	11	1312	13.1	1337	12.9	1427	13.4	1701	15.6	1669	15.3	1772	16.1	1990	17.8
Leukemia	11,858	727	7.8	737	7.6	857	8.8	901	9.1	931	9.3	987	9.9	1021	9.9	1070	10	1175	10.8	1113	10.2	1143	10.4	1196	10.7
Stomach	9764	777	8.3	750	7.8	778	8	840	8.5	861	8.6	813	8.1	841	8.1	837	7.8	828	7.6	836	7.7	798	7.3	805	7.2

Total events: total number of events (2004–2015); *N*: number of diagnoses of a specific cancer; annual specific cancer incidences (*I*): the number of new events divided by the total number of insured persons (at risk population) in the index year. Unit of incidence rate: per 10^5^ cases.

**Table 3 ijerph-16-01982-t003:** Crude rates and direct standardized incidence rates (SIRs) of first-ever cancer diagnosis among insured workers in Taiwan in 2010 in the top thirty industries, stratified by sex.

Industry	All			Male			Female		
	Crude Rate	SIR	Rank	Crude Rate	SIR	Rank	Crude Rate	SIR	Rank
Sand, stone, and clay industry	1401.9	1023.1	1	1142.9	864.3	1	2857.1	1000.0	1
Petroleum and coal products manufacturing	372.7	588.1	2	400.5	706.7	2	230.7	92.4	30
Real estate development	259.9	400.2	3	210.0	476.2	3	309.3	217.0	20
Chemical material manufacturing	227.3	320.9	4	214.9	312.2	10	352.0	249.0	14
Fisheries	460.6	312.2	5	485.8	367.8	4	440.0	283.7	9
Wood and bamboo products manufacturing	398.4	307.7	6	337.8	242.7	25	517.3	353.5	6
Human resources and supply industry	153.7	299.5	7	117.3	228.1	27	196.4	1052.9	2
Creative and artistic performance industry	376.2	295.4	8	424.9	248.6	23	349.6	358.3	5
Construction industry	499.0	285.5	9	531.2	365.6	5	458.0	215.4	22
Unclassified other services	398.4	280.4	10	426.7	326.4	9	392.5	259.6	13
Waste removal, treatment, and recycling	301.0	279.2	11	287.5	311.4	11	345.0	198.1	23
Travel and related booking services	257.7	277.6	12	258.4	302.5	12	259.0	182.3	26
Manufacturing of leather, fur, and their products	362.4	277.3	13	335.3	336.3	6	406.5	260.0	12
Garment and apparel manufacturing	459.1	277.0	14	412.0	222.7	28	484.1	305.3	8
Rubber products manufacturing	182.4	276.8	15	212.6	330.7	7	222.5	223.2	19
Agriculture and animal husbandry	419.7	268.7	16	469.4	282.3	15	389.3	278.2	10
Educational service industry	236.3	266.4	17	238.0	263.0	19	241.2	195.3	24
Pharmaceutical and medical chemical manufacturing	225.8	264.4	18	231.6	148.8	29	229.8	420.5	3
Basic metal manufacturing	220.3	261.6	19	259.3	242.5	26	217.1	333.8	7
Electricity and gas supply industry	279.7	258.7	20	259.7	266.7	17	392.0	170.1	27
Transportation auxiliary industry	386.7	258.5	21	364.0	288.5	13	410.8	233.9	17
Food manufacturing	349.7	256.6	22	357.9	281.9	16	368.8	243.7	15
Printing and data storage media reproduction industry	266.8	250.1	23	229.0	260.8	20	335.4	265.1	11
Accommodation service industry	171.3	243.9	24	134.2	326.9	8	202.1	166.3	28
Land transportation industry	403.4	242.9	25	416.4	250.0	22	321.0	183.6	25
Specialized construction industry	320.8	241.2	26	300.3	265.7	18	379.6	215.5	21
Legal and accounting services	278.2	240.9	27	202.8	144.3	30	310.5	368.6	4
Retail industry	223.8	240.8	28	207.8	257.9	21	240.9	227.2	18
Wholesale industry	205.6	238.4	29	184.5	243.3	24	235.1	236.0	16
Computer, electronics, and optical products manufacturing	125.6	235.6	30	88.3	285.7	14	197.5	127.0	29

**Table 4 ijerph-16-01982-t004:** Characteristics of insured workers with first-time cancer diagnosis in Taiwan, 2004–2015.

Demographic Variables	Employees with a First-Time Cancer Diagnosis	
	*n* = 332,575 (1,564,593 Person-Years)	
	Mean	SD
Age at time of cancer diagnosis	49.7	10.2
Age at time of death	54.3	10.1
Duration from diagnosis to death (number of deaths: 110,692)	2.1	2.2
Length of follow-up for survivors (number of survivors: 221,883)	6.0	3.4
	***n***	**%**
Sex		
Male	154,486	46.5
Female	178,025	53.5
Age at time of cancer diagnosis		
16–39	53,612	16.1
40–44	40,211	12.1
45–49	58,068	17.5
50–54	68,740	20.7
55–59	60,619	18.2
60–64	35,505	10.7
65 and older	15,810	4.8
Year of cancer diagnosis		
2004	22,022	6.6
2005	22,867	6.9
2006	23,857	7.2
2007	25,271	7.6
2008	26,161	7.9
2009	26,777	8.1
2010	28,269	8.5
2011	29,800	9.0
2012	31,343	9.4
2013	31,223	9.4
2014	32,094	9.7
2015	32,891	9.9
Duration from diagnosis to death		
<2 years	71,773	21.6
2–5 years	27,230	8.2
>5 years	11,689	3.5
Survivors	221,883	66.7
Length of follow-up for survivors		
<2 years	27,917	8.4
2–5 years	71,146	21.4
6–10 years	86,002	25.9
>10 years	36,818	11.1

**Table 5 ijerph-16-01982-t005:** Change of employment status within at least 5 years of diagnosis among insured workers with a first-ever cancer diagnosis in Taiwan ^a^.

Demographic Variables	Employees with a First-Time Cancer Diagnosis (*n* = 158,056)	
	*n*	%
Wage or position change at original company		
Yes	44,987	28.5
No	113,069	71.5
Resumption of work (at original company)		
No	5745	3.6
Yes	152,311	96.4
Resumption of work (left original company)		
No	127,486	80.7
Yes	30,570	19.3
Cessation of work (patients who had survived for five years and not worked during this period)		
Still working	139,522	88.3
Did not resume work	18,534	11.7
Death status		
Alive	93,549	59.2
Deceased	64,507	40.8
	**Mean**	**SD**
Resumption of work at original company (number of days)	117.6	190.4
Resumption of work at a different company (number of days)	95.5	227.2
Cessation of work (number of days)	763.0	541.5
Wage range (NT$)	27,729.8	9877.2

^a^ Only employees with a first-ever cancer diagnosis from 2004–2010 were selected to ensure having at least a 5-year follow-up period of labor insurance data for each employee.

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
