# Peer review of "Health Disparities of Employees in Taiwan with Major Cancer Diagnosis from 2004 to 2015: A Nation- and Population-Based Analysis"

_ijerph, 2019, doi:10.3390/ijerph16111982_

Round 1

Reviewer 1 Report

In this study, the authors analyzed the labor insurance and national health insurance research databases with sufficient case number. However, the study design or goal was not interesting. The trends of cancer incidence (figure 1) is just like the trends in Taiwan which indicated that there is no novelty in this investigation. It would be more meaningful to analyze the difference of types of cancer risk between works and non-workers. Furthermore, the differences between types of workers. Another concern is the interpretation of figure 2 which the percentage is not clear defined. It seems that the percentage is divided via the total base number not the workers number of specific jobs? If so, the results should be further analyzed.

Author Response

Comments and Suggestions for Authors

In this study, the authors analyzed the labor insurance and national health insurance research databases with sufficient case number. However, the study design or goal was not interesting. The trends of cancer incidence (figure 1) is just like the trends in Taiwan which indicated that there is no novelty in this investigation. It would be more meaningful to analyze the difference of types of cancer risk between works and non-workers. Furthermore, the differences between types of workers. Another concern is the interpretation of figure 2 which the percentage is not clear defined. It seems that the percentage is divided via the total base number not the workers number of specific jobs? If so, the results should be further analyzed.

Response: Thank you for your comments. In this study, the Labor Insurance Database was used, and this database linked to the National Health Insurance Research Database. Our studied population was the work-related employees. Non-workers may not be obtain information.

For clarity comparison of direct standardize incidence rates (SIR) of employees with first-time cancer diagnosis in Taiwan for the top thirty industries, the differences between types of workersthe were showed as a new Table 5 (p 8) and discussion (p 10). 

Reviewer 2 Report

Major comments: Because the age and sex distributions differ across job sectors, direct comparison of crude incidence rate may be misleading. Using the age and sex standardized rate based on a reference population is more meaningful.  Along the same line, Figure 2 results are not meaningful. Since the base population in each industry type differ, an industry employ most people may have a higher percentage of cancer cases than an industry with a smaller population size. 

Introduction: Introduction should add some background of why the authors pick to examine cancer incidences by occupational types.

Methods: Add some brief explanations of the database used and the coding system used in the database.

In the data sources, there is no mentioning of cancer registry, while in section 2.3., it was mentioned that cancer registry report files are used. Is the cancer registry the 3rd set of data besides the LID and NHIRD being used in the study? Please clarify the linkage among the data sources used. Sample size in each step of the population selection should be given including excluded sample size.

Change “Cases with a behavior code of 2 or 3 in the ICDO-3 are included in the registry.” To ““Cases with a behavior code of 2 (in situ) or 3 (malignant) in the ICDO-3 are included in the registry.”

Results: Did the age and sex distribution across job sectors stay more or less the same during the stud period? How would these changes affect the estimate of cancer incidence rate?

The introduction mentioned that between 43% and 57% of cancers were diagnosed among working age populations in Europe and US. Did the author found similar thing from their data set?

Discussion: The discussion should focus more on the comparisons of cancer incidence by occupational groups, and time trend, as well as whether any cancer prevention and treatment measures may have affect the trend.

The authors mentioned that “Our analysis of first-time cancer diagnoses showed an average length of follow-up for survivors of 6 years and that these survivors had an average salary of 27,730, suggesting that some diagnoses in Taiwan may be attributed to lower economic levels.”. This analysis is too simplistic, as the salary levels vary by many factors including job types and years of worked. The authors appear to have sufficient data to conduct more detailed analysis to identify disparity patterns with regard to economic factors.

Minor comments:

Tables 1-1 and 1-2 can be combined into one table using landscape layout for better viewing. Same for Tables 2-1 and 2-2.

Figure 1. The legend for cancer types is incomplete.

Author Response

Comments and Suggestions for Authors

Major comments:

1.Because the age and sex distributions differ across job sectors, direct comparison of crude incidence rate may be misleading. Using the age and sex standardized rate based on a reference population is more meaningful. Along the same line, Figure 2 results are not meaningful. Since the base population in each industry type differ, an industry employ most people may have a higher percentage of cancer cases than an industry with a smaller population size.

Response: Thank you for your comment. In figure 2, we had presented the age and sex standardized rate based on a WHO reference population in 2000. For clarity comparison of direct standardize incidence rates (SIR) of employees with first-time cancer diagnosis in Taiwan for the top thirty industries, the data has been change as a new Table 5 (p 8).

Introduction:

2.Introduction should add some background of why the authors pick to examine

  cancer incidences by occupational types.

Response: Thank you for your comment. The introduction have been added to ILO and EU explanation that important for monitoring of occupational exposure to carcinogens policy (p 2).

Methods:

3. Add some brief explanations of the database used and the coding system used in the database.

Response: Thank you for your comment. Database and the coding system used have been added to explanations in section 2.2 (p 3).

4.In the data sources, there is no mentioning of cancer registry, while in section 2.3., it was mentioned that cancer registry report files are used. Is the cancer registry the 3rd set of data besides the LID and NHIRD being used in the study? Please clarify the linkage among the data sources used. Sample size in each step of the population selection should be given including excluded sample size.

Response: Thank you for your comment. A description of the cancer registry has been added in section 2.1. Sample size in each step description has been added in section 2.3 (p 3).

5.Change “Cases with a behavior code of 2 or 3 in the ICDO-3 are included in the registry.” To ““Cases with a behavior code of 2 (in situ) or 3 (malignant) in the ICDO-3 are included in the registry.”

Response: Thanks for your comment. We have corrected in section 2.2, accordingly as “Cases with a behavior code of 2 (in situ) or 3 (malignant) in the ICDO-3 are included in the registry” (p 3).

Results:

6.Did the age and sex distribution across job sectors stay more or less the same during the stud period? How would these changes affect the estimate of cancer incidence rate?

Response: Thanks for your comment. A description of age and sex distribution affect the estimate of cancer incidence rate has been added that the age and sex standardized rate based on a WHO reference population. New table 5. showed comparison of direct standardize incidence rates (SIR) of employees with first-time cancer diagnosis stratified by sex, 2010 (p 8).

7.The introduction mentioned that between 43% and 57% of cancers were diagnosed   

  among working age populations in Europe and US. Did the author found similar  

  thing from their data set?

Response: Thanks for your comment. The introduction have been added to related the number of cancer diagnosed at working age population and International Agency for Research on Cancer (IARC) common revision carcinogenic agent standards in Taiwan (p 2).

Discussion:

8. The discussion should focus more on the comparisons of cancer incidence by occupational groups, and time trend, as well as whether any cancer prevention and treatment measures may have affect the trend.

Response: Thanks for your comment. The discussion have been added to related

assessment and comparing concerned to epidemiological studies on occupational risks by sex (p 10).

9. The authors mentioned that “Our analysis of first-time cancer diagnoses showed an average length of follow-up for survivors of 6 years and that these survivors had an average salary of 27,730, suggesting that some diagnoses in Taiwan may be attributed to lower economic levels.”. This analysis is too simplistic, as the salary levels vary by many factors including job types and years of worked. The authors appear to have sufficient data to conduct more detailed analysis to identify disparity patterns with regard to economic factors.

Response: Thanks for your comment. The salary levels have been added to median monthly salary distinguish cancer survivors economic levels (p 11).

Minor comments:

10. Tables 1-1 and 1-2 can be combined into one table using landscape layout for better viewing. Same for Tables 2-1 and 2-2.

Response: Thanks for your comment. We have corrected it. Please see the Table1 and Table 2.

11. Figure 1. The legend for cancer types is incomplete.

Response: Thanks for your comment. We have corrected it.

Round 2

Reviewer 1 Report

The authors answered the question appropriately.

Author Response

Dear Reviewer,

The responses are  attached below, Thanks!

Reviewer 2 Report

The data analysis section is too simplistic and not informative. Please add descriptions of statistic methods used. Some of the analytical methods were mentioned briefly in the footnotes, but need to be detailed in the methods section. For example, how was the direct standardized rate calculated? Please add explanations. While the authors explained in response to the reviewer s’ comments, the detailed and important information need to be added in the manuscript.

The authors should address the limitation of excluding those who left the workforce after cancer diagnosis, and those who are not insured, e.g. potential bias in estimate the overall cancer burden related to occupation, not representing of the entire workforce.

The limitation point, “First, new diagnostic technology and screening methods will yield improvements in cancer data quality in the future, including more accurate and complete data from Taiwan.” appears to imply the quality of the estimate in the current analysis. How do the authors reconcile this conflict?

The discussion should avoid repeating the detailed numbers of results already presented in the Result section; rather highlight/summarize/rephrase the results and discuss their implications.

Please revise the English language throughout the paper. Sentences, such as

“According to statistics, the median recurrent monthly salary was NT$33,502 in Taiwan.1 These results defined those with a salary of less than 70% of that as having lower socioeconomic status.”,

“The assessment conducted in this study was concerned with epidemiological data regarding occupational risks”,  

“As far as we know, this is the first longitudinal study that provides important information in health disparities about the incidence and trends of employees with a first-time cancer diagnosis using data for two populations taken from national labor and health insurance databases in Taiwan.” (what are the two populations? You only have one study population).

… etc. (there are many examples of awkward and imprecise sentences throughout the paper)

need to be revised and improved.

Replace “interference factors” to “confounding and/or mediating factors”

Please revise the table and figure titles as well, and be consistent with the phrases used, for example:

 “Table 1. Number of insured workers, individuals who had been diagnosed with cancer, and individuals with first-time cancer diagnoses, 2004-2015.” to “Table 1. Overall incidence of first-time cancer among insured workers in Taiwan, 2004-2015.”

“Table 2. The cases number and cancer incidences of first-time cancer diagnoses in Taiwanese workers for the top ten cancers, 2004-2015.” to “Table 2. Annual incidence of top ten cancer types among insured workers with first-time cancer diagnosis in Taiwan, 2004-2015.”

“Table 3. The distribution of employees with first diagnosed with cancer in Taiwan from 2004-2015.” to “Table 3. Characteristics of insured workers with first-time cancer diagnosis in Taiwan, 2004-2015.”

“Table 4. The job change of employees with first diagnosed with cancer in Taiwan”

to “Table 4. Employment status change among insured workers after first-time cancer diagnosis in Taiwan, 2004-2015”.

“Figure 1. Trends in annual incidence rates of first-time cancer diagnoses for top seven cancers, 2004–2015.” To “”“Figure 1. Temporal trends in annual incidence of the top seven cancer types among insured workers with first-time cancer diagnosis in Taiwan, 2004–2015.”

Author Response

Dear Reviewer,

The responses are attached below, thanks!

Round 3

Reviewer 2 Report

No further comments.